# EMind: A Foundation Model for Multi-task Electromagnetic Signals Understanding

## Abstract

Deep understanding of electromagnetic signals is a prerequisite for electromagnetic space intelligence applications. Electromagnetic signals, with their high heterogeneity and prominent time-frequency dynamics, pose challenges for extracting representative features. Moreover, the scarcity of high-quality datasets further hinders the performance of electromagnetic signal foundation models under cross-task and cross-scenario conditions. To address these issues, we present a pre-training pipeline for electromagnetic signals foundation model, featuring the creation of a large-scale electromagnetic signal dataset and a foundation model to extract generalizable representations across diverse signals. We curate **EMdata-81M**, a high-quality electromagnetic signal dataset integrating 14 public and in-house sources, which is cleaned, annotated, and formatted uniformly. EMdata-81M comprises 81 million samples, covering 4 scene types, with signal lengths ranging from 128 to 4,096. To enable efficient training of large-scale variable-length data, we propose **EMind**, a foundation model tailored for electromagnetic signals. Specifically, EMind leverages a low-redundancy length adaptive multi-signal packing method and a hardware-aware adjustable dataset weighting strategy, improving representative feature extraction and, in turn, enhancing performance across downstream tasks. Extensive experiments demonstrate that EMind achieves state-of-the-art in various tasks, including modulation classification, parameter regression, radio frequency fingerprinting, and interference recognition, under full fine-tuning, strict train-test splits, and few-shot scenarios. It further attains competitive results on generative tasks including blind source separation and signal denoising. This highlights the effectiveness and scalability of our pipeline in unified electromagnetic signal understanding. The dataset and evluation code is available at: EMind.

## 1 Introduction

Living in a world surrounded by invisible yet ubiquitous electromagnetic (EM) signals, we encounter rich information used in essential areas like communication, navigation, and security. Understanding EM signals, encompassing wireless signal recognition, target parameter prediction, interference identification and waveform denoising and reconstruction, underlies cornerstones of critical applications such as cognitive radio technology and integrated sensing and communication (ISAC).

Inspired by the breakthroughs in foundation models in computer vision Dosovitskiy et al. (2020) and speech processing Radford et al. (2023), the study on EM signal foundation models has gained attention Hao et al. (2023); Zhang et al. (2023b). However, EM signals differ fundamentally from images, videos, and audio. They exhibit more significant data distribution shifts than images, extreme sparsity compared to videos, and stronger non-stationary properties than audio. Moreover, EM signals manifest not only diverse types with high heterogeneity but also prominent time-frequency dynamics (see Figure 1 (a)). These characteristics hinder the direct transfer of existing models, requiring the development of specialized foundation models for this modality. The development of EM signal foundation models faces two major issues: (i) most EM signals come from from non-cooperative scenarios with encrypted or undisclosed protocols, leading to fragmented and non-standardized data, the difficulty of acquiring high-quality datasets impedes the sufficient training of foundation models; (ii) existing datasets Chi et al. (2024) Chen et al. (2024) Chen et al. (2025) Zhou et al. (2025) lack

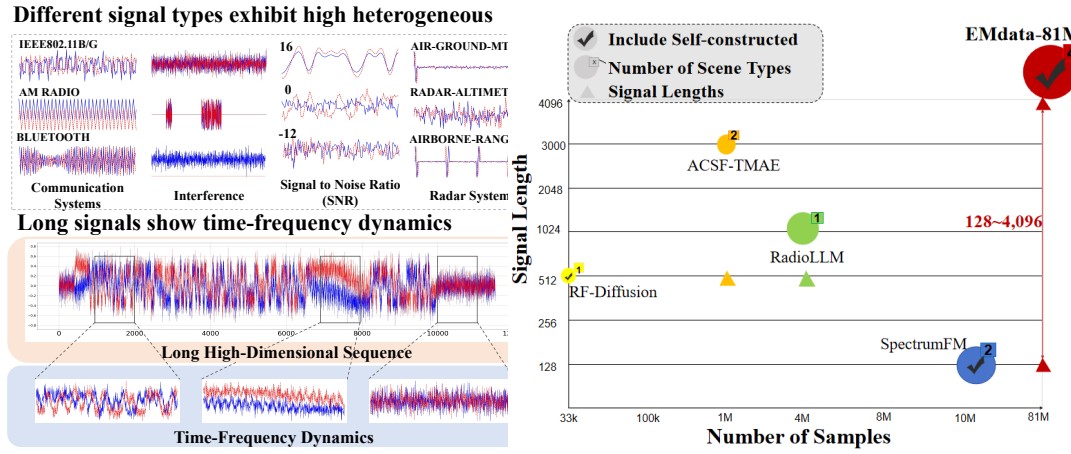

(a) Visualization of signals        (b) Comparison of EM signals pre-train datasets

Figure 1: (a) Challenge of processing EM signals, including heterogeneity and time-frequency dynamics. (b) Comparison of EM signals pre-train datasets: the volume of EMdata-81M surpasses at least 5× compared to existing datasets, encompassing a broader range of scene types from both public and in-house sources, and covering a wider span of signal lengths.

diversity in scene types and signal lengths, with limited data scale, restricting the generalization of learned representations.

In address this challenge, we first systematically integrated EM signal datasets from various scene types, including communication, radar, radio frequency (RF), and interference. However, due to multiple data sources, varying formats, and inconsistent annotations, we conducted cleaning and standardized processing. All data are unified into raw IQ (In-Phase/Quadrature) format with parameters annotated, which leads to the creation of the largest known EM signal dataset, **EMdata-81M**, containing 81 million signal samples. The dataset is designed to fully leverage the representation learning potential of large-scale pretrain-finetune paradigms for EM signal applications. EMdata-81M surpasses previous datasets Chi et al. (2024) Chen et al. (2024) Chen et al. (2025) Zhou et al. (2025) by at least five times (see Figure 1 (b)), encompassing diverse requirements for downstream tasks, including various transmitters and receivers, multiple signal types, and a wide range of signal lengths and sampling rates.

Despite recent advances in EM signal foundation models Chen et al. (2025) Zhou et al. (2025), large-scale pre-training remains hindered by heavy computational demands and slow convergence. In particular, when scaling up to EMdata-81M, two key challenges emerge: (i) signal lengths vary dramatically from 128 to 4,096 samples (a 32× difference), complicating efficient training; and (ii) data from diverse sources require control of weighting to ensure balanced and sufficient training. To handle variable lengths, we adopt low-redundancy length adaptive multi-signal packing, aligning 128 to 4,096 sample signals per batch to reduce padding and ensure unbiased gradients. For dataset imbalance, we employ hardware-aware adajustable dataset weighting strategy, down-weighting overfitted datasets and up-weighting harder ones for balanced coverage and stable convergence in 80M-scale training. Building on these techniques, we propose **EMind**, a foundation model tailored for EM signals, the learned representations advance multiple downstream tasks including Automatic Modulation Classification (AMC), Radar Waveform Classification (RWC), Radar Parameter Estimation (RPE), Wireless Interference Identification (WII), and Radio Frequency Fingerprinting Identification (RFFI). More importantly, the representations empower generative models, benefiting intricate reconstruction tasks like Blind Source Separation (BSS) and Signal Denoising (SD). The proposed EMind demonstrates both the broad applicability of pre-training in EM tasks and the potential to advance EM intelligence from task-specific models toward general-purpose understanding (see Fig. 2). In summary, this paper presents three main contributions:

- We construct EMdata-81M, the largest EM dataset to date, with 81 million samples covering diverse signal types, rich attribute labels, and variable sample lengths, all standardized stored to provide a rubost basis for large-scale pre-training.

- We propose EMind, a foundation model specifically tailored for EM signals. The network architecture of EMind is designed for the EM modality, employing low-redundancy length

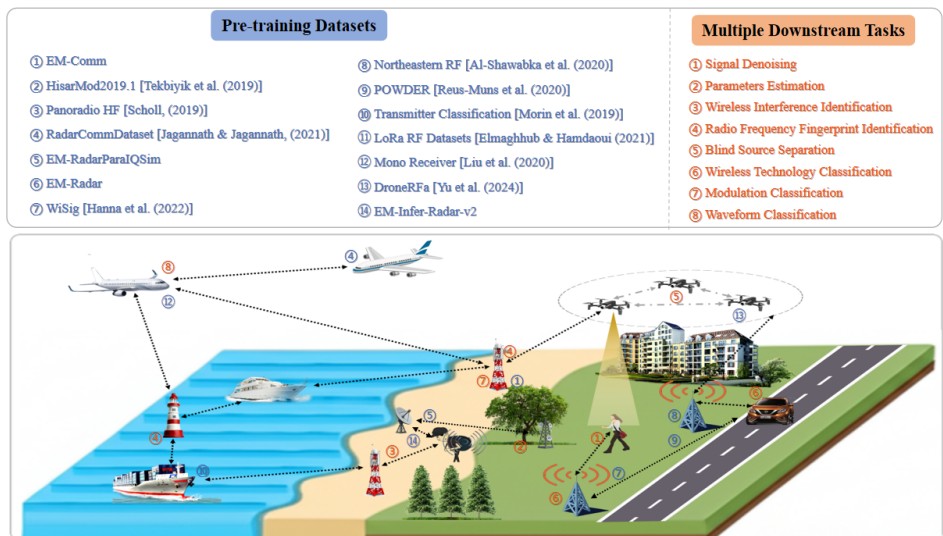

Figure 2: EMind: A foundation model for EM signals, capable of multitask learning and applicable across diverse domains such as communication, navigation, and security.

    adaptive multi-signal packing and hardware-aware adjustable dataset weighting strategy, which ensures the acquisition of generalizable and transferable representations.

- EMind systematically validates the applicability of pretrain–finetune paradigm for EM signal tasks, markedly reducing reliance on task-specific designs and setting new SOTA results. Notably, EMind effectively extends to generative tasks like BSS and SD, empirically showing that learned features retain their efficacy thus broadening the paradigm's applicability in the EM signal understanding.

## 2 PROBLEM FORMULATION AND CHALLENGES

EM signals are a special type of time-series data, typically transmitted through the wireless channel. At the receiver, a real-valued signal $x(t)$ with carrier frequency $f_c$ can be witten as,

$$x(t) = I(t)\cos(2\pi f_c t) - Q(t)\sin(2\pi f_c t), \tag{1}$$

where $I(t)$ and $Q(t)$ represent the in-phase and quadrature-phase components of the signal, respectively. The baseband signal is discretized at a sampling rate $f_s$ and digitally downconverted (DDC), yielding a discrete complex IQ signal, which is,

$$s[n] = I[n] + jQ[n], \quad n = 0, 1, 2, \ldots, N-1. \tag{2}$$

Here, $N$ represents the length of the samples. As the complexity of the EM space continues to increase, both the types and quantity of EM signals are rapidly growing. However, due to non-negligible differences in signal features and task types, single-task models show poor generalization and low training efficiency. Therefore, there is an urgent need for a foundation model capable of learning generalizable representations for multiple downstream tasks. Nevertheless, developing such an EM signal foundation model still faces the following challenges:

**Challenge 1: Constructing a unified pre-training dataset for EM signals.** Current EM signal datasets face several bottlenecks. Simulation data coarsely model complex real-world conditions, limiting their ability to replicate realistic EM environments. Self-collected data are scarce due to non-cooperative sources and encrypted or undisclosed protocols, resulting in missing key information and few labelable samples. Open-source datasets are limited in size and diversity, with inconsistent metadata and file formats, making cross-dataset integration costly. Hence, obtaining diverse, large-scale, high-quality pre-training dataset for EM signals remains challenging, forming a key bottleneck for foundation model development in this area.

**Challenge 2: Developing an effective foundation model toward EM signals.** EM signals are highly diverse, with heterogeneous characteristics and propagation traits, and include both long-term sequences and short bursts, each requiring different training preferences. Therefore, designing a network architecture that accounts for the physical properties of EM signals while effectively extracting their universal features, has become the core challenge in building EM foundation models.

# 3 DATASET CURATION

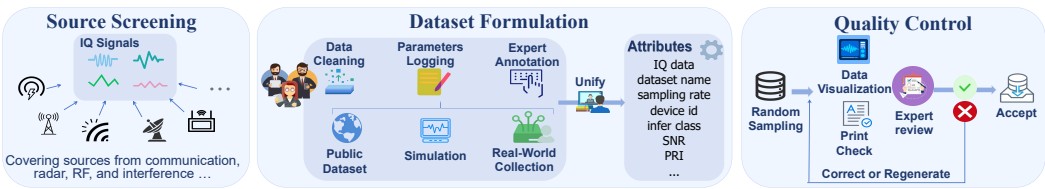

Figure 3: Pipeline of EM pre-train dataset cosntruction. Our pipeline comprises 3 stages all led by experts: Source Screening, Dataset Formulation, and Quality Control.

Table 1: Details of EMdata-81M. Bold denotes self-constructed datasets.

| DATASET NAME | TRANSMITTER | RECEIVER | SIGNAL TYPE | SIGNAL LENGTH | SAMPLING RATE | DATA* SOURCE | NUMBER OF SAMPLES |
|---|---|---|---|---|---|---|---|
| **EM-Comm** | **a USRP X310** | - | **Category 14 modulated signals, SNR: -20~18db** | **1,024** | **20 MHz** | **Real-world** | **2,747,000** |
| HisarMod2019.1 Tekbıyık et al. (2019) | - | - | Category 26 modulated signals, SNR: -20~18db | 1,024 | 1 MHz | Simulation | 780,000 |
| Panoradio HF Scholl (2019) | - | - | Category 18 modulated signals, SNR: -10~25d | 2,048 | 6 kHz | Simulation | 172,800 |
| RadarCommDataset Jagannath & Jagannath (2021) | a USRP N210 | USRP N210 | Category 6 modulated signals, Category 8 signal types, SNR: -20~18db | 128 | 10 MHz | Real-world &Simulation | 860,361 |
| **EM-RadarParaIQSim** | - | - | **Category 2 radar types, Category 3 parameters, SNR: 6~12db** | **320/384 400/480** | **10/12 MHz** | **Simulation** | **300,000** |
| **EM-Radar** | - | - | **10 radar emitters, including 5 radar waveforms, and 3 radar system parameters SNR: -10~20db** | **1,024** | **5 MHz** | **Simulation** | **327,680** |
| WiSig Hanna et al. (2022) | 174 WiFi Emitter | 41 USRP B210/N210/X310 | IEEE802.11a/g | 256 | 25 MHz | Real-world | 18,243,630 |
| Northeastern RF Al-Shawabka et al. (2020) | 20 USRP devices | - | IEEE802.11a/g | 288 | 20 MHz | Real-world | 17,930,880 |
| POWDER Reus-Muns et al. (2020) | 4 USRP X310 | USRP B210 | 5G, WiFi, LTE | 512 | 5/7.68 MHz | Real-world | 1,044,472 |
| Transmitter Classification Morin et al. (2019) | 21 USRP devices | USRP N2932 | IEEE 802.15.4 | 600 | 5 MHz | Real-world | 11,928,835 |
| LoRa RF Datasets Elmaghbub & Hamdaoui (2021) | 25 Pycom devices | USRP B210 | LoRa | 4,096 | 1 MHz | Real-world | 13,498,730 |
| Mono Receiver Liu et al. (2020) | Over 140 civil aircrafts | USRP B210 | ADS-B | 974 | 8 MHz | Real-world | 30,367 |
| DroneRFa Yu et al. (2024) | 24 categories of UAVs | USRP | Communication RF | 1,024 | 100 MHz | Real-world | 13,132,770 |
| **EM-Infer-Radar-v2** | - | - | **Category 12 radar interference types ISR: 30~60db** | **2,048** | **20 MHz** | **Simulation** | **120,000** |
| TOTAL | | | | | | | 81,117,525 |

As show in Figure 3, we integrate multi-source datasets from communications, radar, RF, and interference, all stored in raw IQ format. To ensure quality for large-scale pretraining, the data is rigorously cleaned, annotated by experts, and converted to an n×2 Parquet format. Data quality is further verified via random sampling and visual inspection. The above mentioned process yields the largest known EM signal dataset, EMdata-81M. **Source Screening.** We construct EMdata-81M for a diversity in signal types and tasks, ensuring comprehensive and effective training data. It integrates 14 datasets, combining public sources with self-collected data to fill gaps. As shown in Table 1, EMdata-81M covers various devices, scene types, and signal lengths from 128 to 4,096 with sampling rates of 6 kHz–100 MHz. Public datasets contribute 77,622,845 samples, while 4 self-collected datasets (bolded, prefixed EM-) add 3,494,680, all rigorously selected and pre-processed. In total, EMdata-81M contains 81,117,525 samples, making it the largest known EM signal pre-training dataset. **Expert Annotations.** For annotation, public datasets use publisher-provided labels; simulation datasets generate la-

Table 2: Main statistics of EMdata-81M.

| STATISTIC | NUMBER OF SAMPLES |
|---|---|
| **Task type** | |
| - Communication | 3,699,800 |
| - Radar | 1,488,041 |
| - RF | 75,809,684 |
| - Interference | 120,000 |
| **Attributes** | |
| - Modulation | 4,560,161 |
| - Radar Waveform | 627,680 |
| - Signal-to-Noise Ratio (SNR) | 5,218,208 |
| - Interference class | 120,000 |
| - Interference-to-Signal Ratio (ISR) | 120,000 |
| - Band width | 327,680 |
| - Pulse repetition interval (PRI) | 627,680 |
| - Pulse width | 627,680 |
| - Device id | 75,809,684 |
| - Transmission id | 17,930,880 |
| **Signal Length** | |
| - 128 | 860,361 |
| - 256 | 18,243,630 |
| - 288 | 17,930,880 |
| - 320 | 75,000 |
| - 384 | 75,000 |
| - 400 | 75,000 |
| - 480 | 75,000 |
| - 512 | 1,044,472 |
| - 600 | 11,928,835 |
| - 974 | 30,367 |
| - 1,024 | 16,987,450 |
| - 2,048 | 292,800 |
| - 3,840 | 12,399 |
| - 4,096 | 13,498,730 |

bels automatically from predefined modulations and parameters; self-collected datasets derive labels from recorded information such as device type, distance, frequency, modulation, and signal strength. **Quality Control.** A quality control process ensures data integrity and task relevance. IQ samples are randomly checked, parameters printed, and waveforms plotted; anomalies in padding, format, or content are flagged and corrected by experts, maintaining dataset reliability.

Table 3: Comparison of pre-train datasets.

| PRRE-TRAIN DATASET | DATASET SOURCE VOLUMN | INCLUDE SELF-CONSTRUCTED | SIGNAL LENGTH | SCENE TYPE | NUMBER OF SAMPLES |
|---|---|---|---|---|---|
| RF-Diffusion Chi et al. (2024) | 2 | ✓ | 512 | RF | 33 k |
| ACSF-TMAE Chen et al. (2024) | 4 | ✗ | 128 & 3,000 | communication & RF | 1 million |
| RadioLLM Chen et al. (2025) | 4 | ✗ | 128 & 1,024 | communication | 4 million |
| SpectrumFM Zhou et al. (2025) | 3 | ✓ | 128 | communication & radar & RF | 8-10 million* + 25 GB† |
| EMdata-81M | 14 | ✓ | 128~4096 | communication & radar & interference | 81 million |

* In Zhou et al. (2025), the pre-training data volume for TechRec is estimated according to https://github.com/JaronFontaine/Technology-Recognition-dataset-of-real-life-LTE-WiFi-and-DVB-T?tab=readme-ov-file
† 25GB is a self-collected dataset without length unspecified.

**Analysis.** EMdata-81M stands out in scene types, signal lengths, and dataset scale, covering 4 scene types, 29 modulation types, 9 radar waveforms, and 12 interference classes, with details provided in the appendix. **i) Diversity of task types.** EMdata-81M includes various scene types, such as communication,

radar, interference, and wireless RF datasets, covering protocols and devices from WiFi, LoRa to ADS-B, UAV, and more. **ii) Comprehensive annotations.** We provide detailed attribute annotations including modulation for communication signals, waveform, pulse width, and pulse repetition period for radar, device ID and bandwidth for RF, and interference type and ISR for interference. All samples include sampling rate (kHz–MHz) and SNR (if available). **iii) Length of Samples.** EMdata-81M contains samples ranging from 128 to 4,096 in length, enabling training across a wider range. In contrast, existing pretraining datasets typically cover a single scenario or a few fixed lengths (Table 3). The diversity in signal lengths allows the model to better handle complex signals and improves real-world generalization, with detailed signal length statistics shown in Table 2.

# 4 EMIND: ELECTROMAGNETIC SIGNAL FOUNDATION MODEL

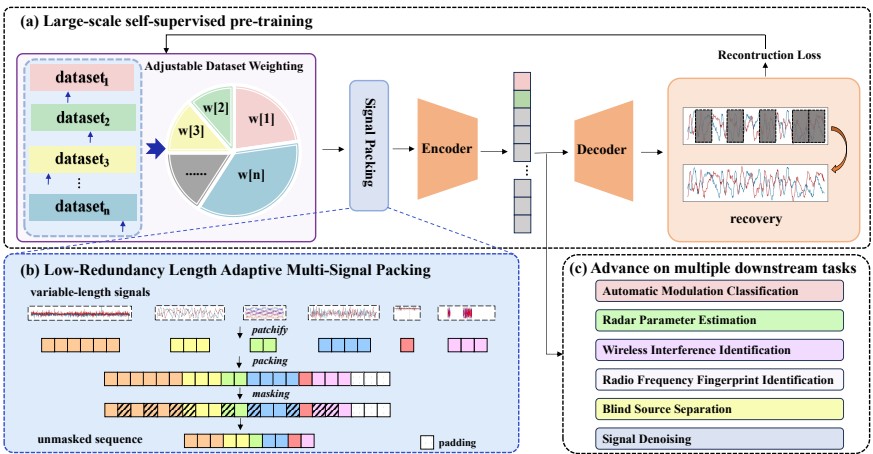

Figure 4: Overview of EMind. (a) Large-scale self-supervised pre-training with adjustable dataset weighting. (b) Low-redundancy length adaptive multi-signal packing and per-sample masking. (c) Adapt to multiple downstream tasks with arbitrary-length signal inference.

To handle the large variation in signal lengths, we propose a packing method that concatenates IQ samples of different lengths into long sequences, enabling low-redundancy multi-length training and inference with arbitrary lengths (Fig. 4(b), Sec. 4.1). To mitigate training imbalance in large-scale diverse datasets, we introduce a hardware-aware framework that adjusts sampling weights, by up-weighting difficult datasets and down-weighting easier ones for stable and balanced training (Fig. 4(a), Sec. 4.2). Building on these techniques and a Transformer encoder-decoder, we adopt a masked autoencoder (MAE) for large-scale multi-scenario pretraining. For downstream fine-tuning, the shared backbone supports multiple downstream tasks (Fig. 4(c)). The proposed EMind demonstrates strong cross-domain generalization, achieving high-quality multi-task EM signal modeling.

## 4.1 LOW-REDUNDANCY LENGTH ADAPTIVE MULTI-SIGNAL PACKING

To enable efficient training by supporting multiple length samples and minimizing redundancy, we propose a length adaptive multi-signal packing method. The training data come from different datasets with sample lengths that vary dramatically, ranging from 128 to 4,096, spanning a 32-times difference. In such a highly non-uniform length distribution, traditional padding methods introduce excessive zero padding, leading to increased memory usage and computational overhead. To address this is-

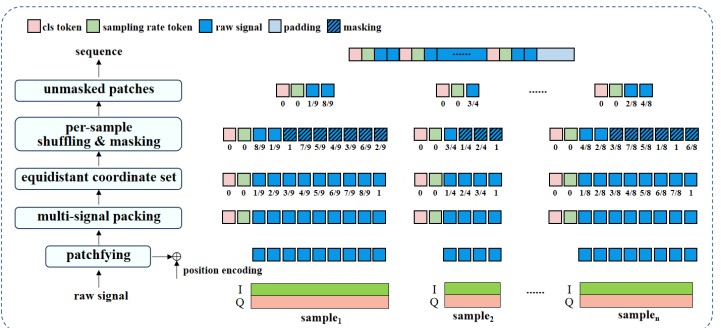

Figure 5: Multiple signal samples are dynamically packed into fixed-length sequences, followed by per-sample masking strategy.

sue, we design a dynamic packing mechanism that uses a predefined sequence length as the packing capacity, filling samples in sequential order. When the remaining capacity is insufficient to accommodate a new sample, a new sequence is initiated. The specific process is shown in Figure 5. The signal packing strategy alleviates computational redundancy and memory bottlenecks, enabling efficient training on large-scale datasets.

After packing, each sequence contains multiple signals of varying lengths, with ambagious sample boundaries. If the entire sequence is masked directly, the masking ratios of different samples within the sequence differ significantly. For example, shorter samples may be completely masked, while longer samples may be insufficiently masked, making precise sample-level masking difficult. To ensure the model distinguishes each signal, the boundary index of each sample is recorded for independent random masking. Traditional loop-based masking controls apply masks to individual samples, but the computational complexity largely increases, reducing the efficiency of sequence packing. To this end, we propose an innovative mask control mechanism based on vectorized operations, eliminating the need for explicit loops. Specifically, for a signal sample of length $l$, we normalize its sample point positions to an equidistant coordinate set in the interval [0,1], represented as $[0, \frac{1}{l-1}, \frac{2}{l-1}, \ldots, 1]$. Using this standardized position marker, we randomly shuffle the order of the sample points to select a fixed ratio of valid patches according to the masking ratio. Therefore, regardless of the signal length, the masked data of each sample is uniform and the masking ratio remains accurate, thus avoiding masking ratio fluctuations.

### 4.2 Hardware-aware adjustable Dataset Weighting

An adjustable dataset sampling weight strategy is employed to tackle with the issue of inconsistent convergence speeds across different datasets during training. Based on empirical prior knowledge, the sampling weight for datasets that are difficult to learn is raised to enhance the model's ability to learn from these datasets, while for datasets that are easier to learn or prone to performance degradation, the sampling frequency is reduced to mitigate the risk of overfitting. To control the sampling weights of different datasets, we develop a hardware-aware training framework based on the producer-consumer mechanism, achieving accurate ratio control and sample packing coordination. The framework is maintained by CPU, which manages a unified buffer for data loading, preprocessing, and cache management, and employs streaming scheduling to allow GPU to alternately perform data input and model computation. Within the buffer, we implement a memory-mapped sequential access strategy, allocating independent pointers for each dataset and adjusting the pointer movement speed according to the preset ratio, ensuring real-time balanced data input and training of multi-level training samples. Detailed pretraining sampling weights for each dataset are provided in the Appendix.

## 5 Experiments

This section provides a brief overview of the pretraining setup, downstream task configurations, and experimental results. Additional data details, model implementation details, more experiments, and visualization results are provided in the appendix.

Table 4: Downstream Multi-task Dataset

| DATASET | TASK TYPE* | CLASS NO. | ATTRIBUTE | RANGE | SIGNAL LENGTH | SAMPLING RATE | DATA SOURCE |
|---|---|---|---|---|---|---|---|
| RML2016.10A O'shea & West (2016) | AMC (Cls.) | 11 | SNR | $-20 \sim 18\,dB$ | 128 | 1 MHz | - |
| RML2016.10B O'shea & West (2016) | AMC (Cls.) | 10 | SNR | $-20 \sim 18\,dB$ | 128 | 1 MHz | - |
| RML2016.04C O'Shea et al. (2016) | AMC (Cls.) | 11 | SNR | $-20 \sim 18\,dB$ | 128 | 1 MHz | - |
| RML2018.01A O'Shea et al. (2018) | AMC (Cls.) | 24 | SNR | $-20 \sim 30\,dB$ | 1,024 | 1 MHz | - |
| ADS-B Ya et al. (2022) | RFFI (Cls.) | 198 | - | | 3,000 | 50 MHz | - |
| | RWC (Cls.) | 5 | SNR | $-20 \sim 20\,dB$ | | | |
| | | | PULSE NMUBER | $2 \sim 6$ | | | |
| RadChar Huang et al. (2023) | RPE (Reg.) | | PULSE WIDTH | $10 \sim 16\,\mu s$ | 512 | 3.2 MHz | - |
| | | | PRI† | $17 \sim 23\,\mu s$ | | | |
| | | | PULSE TIME DELAY | $1 \sim 10\,\mu s$ | | | |
| **EM-AIS** | RFFI (Cls.) | 112 | - | | 3,840 | 156.25 MHz | Real-world |
| **EM-Infer-Comm** | WII (Cls.) | 9 | SIR | $-20 \sim 20\,dB$ | 1,024 | 20 MHz | Simulation |
| **EM-Infer-Radar** | WII (Cls.) | 12 | SIR | $-20 \sim 20\,dB$ | 1,024 | 20 MHz | Simulation |
| **EM-Radar-Mix** | BSS (Rec.) | - | SNR | $12\,dB$ | 1,024 | 5 MHz | Simulation |
| **EM-Signal-Denoise** | SD (Rec.) | - | SNR | $-3 \sim 20\,dB$ | 1,024 | 20 MHz | Simulation |

* AMC represents Automatic Modulation Classification; RWC refers to Radar Waveform Classification; RPE stands for Radar Parameter Estimation; WII denotes Wireless Interference Identification; RFFI indicates Radio Frequency Fingerprint Identification; SD refers to Signal Denoising; and BSS denotes Blind Source Separation.
† PRI denotes for pulse repetition interval.

### 5.1 Main Results

The core feature of foundation models lies in their generalization to diverse downstream tasks. Thereby, we select downstream datasets independent of the pre-training data and evaluated them

on classification, regression, and reconstruction tasks, as shown in Table 4 (self-constructed datasets are denoted with the EM- prefix). We included rich evaluation metrics for different tasks: Overall Accuracy (OA) for classification, Mean Absolute Error (MAE) for regression, Mean Squared Error (MSE), Signal-to-Distortion Ratio (SDR), Signal-to-Interference Ratio (SIR), Signal-to-Artifacts Ratio (SAR), and Scale-Invariant Signal-to-Distortion Ratio (SI-SDR) for blind source separation, and MSE also for denoising. Few-shot classification further introduces Cohen's Kappa Coefficient (Kappa). Detailed metric descriptions are provided in the appendix.

### 5.1.1 CLASSIFICATION

Table 5: Classification results compared with State-of-the-art. BOLD indicates the best performance.

| TASK | AMC | | | | RFFI | | WII | |
|---|---|---|---|---|---|---|---|---|
| METHOD | RML2016.10A* | RML2016.10B* | RML2016.04C* | RML2018.01A | ADS-B | EM-AIS | EM-Infer-Comm | EM-Infer-Radar |
| ResNet He et al. (2016) | 50.04 | 54.88 | 55.97 | 43.06 | 84.51 | 42.02 | 76.79 | 71.64 |
| MCNet Huynh-The et al. (2020) | 53.52 | 59.22 | 59.57 | - | - | - | - | - |
| CNN2 O'Shea et al. (2018) | 53.25 | 57.14 | 59.45 | - | - | - | - | - |
| GRU2 Hong et al. (2017) | 58.80 | 64.11 | 63.13 | - | - | - | - | - |
| DAE Ke & Vikalo (2021) | 58.97 | 61.46 | 55.91 | - | - | - | - | - |
| CGDNN Njoku et al. (2021) | 56.57 | 58.26 | 60.34 | - | - | - | - | - |
| Transformer Vaswani et al. (2017) | 59.27 | 63.10 | 65.41 | 59.45 | 78.77 | 39.12 | 80.44 | 77.05 |
| MSNet Zhang et al. (2021) | 58.33 | 63.49 | 63.66 | - | - | - | - | - |
| AMC_Net Zhang et al. (2023a) | 59.10 | 63.38 | 63.01 | - | - | - | - | - |
| SpectrumFM Zhou et al. (2025) | 63.72 | 65.35 | 73.37 | - | - | - | - | - |
| EMind | **62.51** | **65.45** | **74.34** | **63.83** | **99.87** | **57.07** | **83.11** | **79.19** |

* As stated in the scikit-learn documentation (https://scikit-learn.org/stable/modules/generated/sklearn.metrics.recall_score.html), weighted recall is equivalent to accuracy in single-label multiclass settings. Therefore, we treat the recall reported in Zhou et al. (2025) as overall accuracy (OA) for RML2016.10A, RML2016.10B, and RML2016.04C.

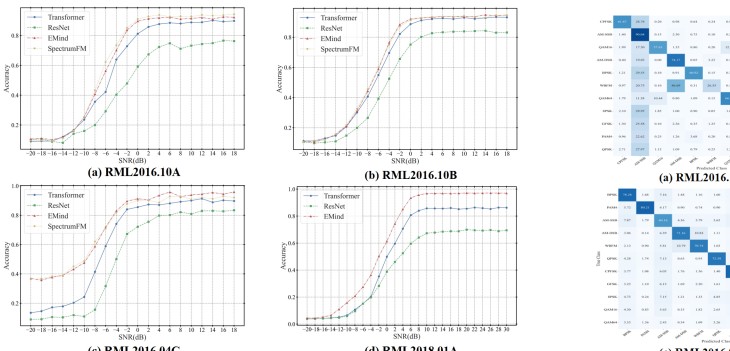

Figure 6: Performance comparison in terms of accuracy at varying SNRs.

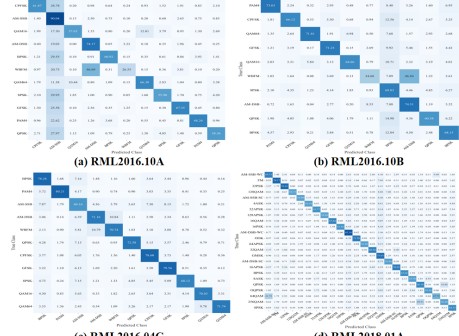

Figure 7: Confusion matrices highlights the performance across various modulations.

**Automatic Modulation Classification (AMC), Radio Frequency Fingerprint Identification (RFFI) and Wireless Interference Identification (WII).** In the classification task, the train/test split for RML2016.10A O'shea & West (2016), RML2016.10B O'shea & West (2016), RML2016.04C O'Shea et al. (2016), and RML2018.01A O'Shea et al. (2018) follows SpectrumFM Zhou et al. (2025) for fair comparison; ADS-B, EM-Infer-Comm, and EM-Infer-Radar use a more strict 1:9 split; and for EM-AIS, a 5:5 split is used to ensure that all classes are covered in the training set. The classification results in Table 5 show that our model outperforms others in AMC, RFFI and WII. Figure 6 compares EMind with other methods on the RML four datasets accross SNRs. Figure 7 displays the confusion matrix of EMind on these datasets, highlighting its classification performance across modulation categories and potential misclassification areas.

Table 6: Classification results for different methods under the 1:9 dataset split. BOLD indicates the best performance, and *italics* indicate the second best.

| METHOD | RML2016.10A* | RML2016.10B* | RML2016.04C* | RML2018.01A |
|---|---|---|---|---|
| ResNet He et al. (2016) | 43.14 | 51.17 | 47.76 | 40.39 |
| Transformer Vaswani et al. (2017) | 43.98 | 53.13 | 48.49 | 52.93 |
| **EMind** | *49.51* | *57.51* | *54.37* | *55.39* |
| **EMind-large** | **50.81** | **58.27** | **55.89** | **57.54** |

Table 7: Few-shot classification results for AMC (RML2016.10A) and RWC (RadChar) tasks.

| | RML2016.10A | | | | RadChar | | | | | |
|---|---|---|---|---|---|---|---|---|---|---|
| | 50 shot | | 100 shot | | 10 shot | | 50 shot | | 100 shot | |
| METHOD | OA(%) | Kappa | OA(%) | Kappa | OA(%) | Kappa | OA(%) | Kappa | OA(%) | Kappa |
| ResNet | 38.97 | 32.87 | 42.01 | 36.21 | 71.94 | 64.93 | 79.89 | 74.86 | 80.97 | 76.21 |
| Transformer | 38.40 | 32.25 | 39.81 | 33.79 | 64.27 | 55.37 | 76.42 | 70.52 | 79.21 | 74.01 |
| **EMind** | **48.38** | **42.85** | **50.10** | **45.12** | **78.37** | **72.97** | **81.94** | **77.43** | **83.14** | **78.93** |

**Challenge data split setup.** Inspired by remote sensing research Wang et al. (2025), we adopt an extreme 1:9 train/test split to valid the generalization ability of the proposed EMind, and introduces EMind-Large (details in appendix) as a comparison. Table 6 shows that, on the four RML datasets, EMind-Large achieves the best overall performance, setting new records for most metrics, and EMind-base is the second best.

**Few-shot.** Two representative datasets, RML2016.10A O'shea & West (2016) for AMC and Rad-Char Huang et al. (2023) for RWC, are used for few-shot experiments. As shown in Table 7, EMind outperforms all methods in every few-shot setting. Notably, the test datasets are strictly separated from the pretraining datasets, highlighting the model's real-world generalization capability. Especially, in the 10-shot setting, RadChar Huang et al. (2023) still achieved competitive performance with 78.37%.

Table 8: Multi-task of RWC and RPE on RadChar Huang et al. (2023). BOLD indicates the best performance. Lower MAE indicating better regression performance, while higher accuracy reflects better classification.

| | MAE (μs) | | | | ACCURACY (%) |
| | Pulse Number | Pulse Width | Pulse Repetition Interval (PRI) | Pulse Time Delay | Radar Waverfrom |
| METHODS | -10db 0db 10db all† | -10db 0db 10db all | -10db 0db 10db all | -10db 0db 10db all | -10db 0db 10db all |
|---|---|---|---|---|---|
| CNN1D | 0.729, 0.193, 0.085  - | 1.413, 0.560, 0.340  - | 0.999, 0.330, 0.209  - | 1.349, 0.385, 0.206  - | 75.7, 99.8, 100  - |
| CNN2D | 0.793, 0.174, 0.090  - | 1.466, 0.801, 0.505  - | 1.054, 0.420, 0.299  - | 1.729, 0.638, 0.443  - | 67.3, 98.3, 99.8  - |
| IQST-S Huang et al. (2023) | 0.733, 0.294, 0.251  - | 1.282, 0.628, 0.364  - | 0.816, 0.273, 0.192  - | 1.229, 0.415, 0.277  - | 79.2, 99.9, 100  - |
| IQST-L Huang et al. (2023) | 0.752, 0.195, 0.124  - | 1.253, 0.579, 0.334  - | 0.799, 0.286, 0.225  - | 1.253, 0.379, 0.233  - | 79.1, 99.8, 100  - |
| **EMind** | **0.330, 0.006, 0.005, 0.114** | **0.797, 0.197, 0.080, 0.305** | **0.463, 0.109, 0.085, 0.221** | **0.708, 0.149, 0.092, 0.323** | **86.65, 100, 100, 88.49** |

† -10 dB, 0 dB, and 10 dB denote results at specific SNR respectively, all denotes results evaluated across all SNR [-20:20] dB.

### 5.1.2 PARAMETERS REGRESSION

For regression task, we achieve multiple parameters regression by extending the regression head into a multi-output structure, and support joint training of regressions and classification. Experiments on the RadChar Huang et al. (2023) dataset involve classifying radar waveforms and predicting four parameters, which are pulse count, pulse width, pulse repetition interval, and pulse time delay (configurations in Table 4). The train/val/test split is 70:15:15 for fair comparison with Huang et al. (2023). Experimental results in Table 8 show the proposed EMind substantially outperforms the current SOTA iacross all classification and regression metrics.

Table 9: Performance Comparison of BSS on EM-Radar-Mix. For SDR, SIR, SAR, and SI-SDR, higher is better; for MSE, lower is better.

| Setting | SDR | SIR | SAR | SI-SDR | MSE |
|---|---|---|---|---|---|
| Linear prob | 4.85 | 10.73 | 6.92 | -5.88 | -13.85 |
| Fine-tune | 5.74 | 11.60 | 7.83 | -3.85 | -15.32 |

### 5.1.3 BLIND SOURCE SEPARATION (BSS) AND SIGNAL DENOISE (SD)

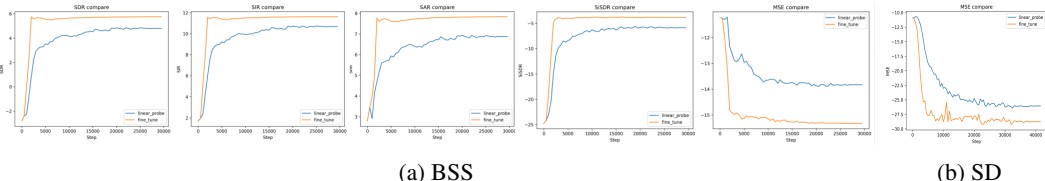

(a) BSS                                          (b) SD

Figure 8: Performance comparison of fine-tune and linear prob on BSS over EM-Radar-Mix and SD over EM-Signal-Denoise.

BSS is a highly challenging inverse problem in complex EM environments, aiming to recover original, mutually independent source signals from observed mixtures without prior knowledge of source characteristics or mixing processes. Only noisy IQ signals from an unknown number of sources through unknown channels are observed, so both the source count and ground-truth signals are unavailable during training. The problem is ill-posed, lacking a unique solution without additional assumptions, and highly sensitive to small input perturbations. Due to the lack of high-quality public datasets, we constructed EM-Radar-Mix (details in the appendix) for BSS with up to two sources. Following the audio BSS Eval toolbox Vincent et al. (2006), we use multiple metrics, and add a signal-specific metric SI-SDR. As shown in Table 9 and Fig. 8a, results for fine-tuning and linear probing demonstrate EMind's superiority in both quantitative performance and training convergence for BSS. Visualization results and more details are found in the appendix.

We also introduce the self-collected dataset EM-Signal-Denoise (see appendix) to evaluate the SD task under the BSS framework, separating clean signals from noise. The framework takes noisy signal as input without supervision of denoised data. Fig. 8b shows MSE curves for linear probing and fine-tuning; visualizations and details are in the appendix.

### 5.2 ABLATION STUDY

To validate the efficacy of EMind, we conducted ablation studies on the packing strategy, mask method, mask ratio, and max sequence length. We selected two representative tasks, AMC on

RML2016.10B O'shea & West (2016) and WII on EM-Infer-Comm, for full fine-tuning and linear probing experiments. All ablation experiments were performed under a more stringent 1:9 training-to-testing ratio.

**Packing Strategy.** Multi-signal packing strategy greatly improved the efficiency of training on large-scale EM data. We conducted an ablation study on the strategy of using or not using packing, and the results are shown in Table 10.

Table 10: Training efficiency comparison of packing strategies.

| | Train Steps | Samples per Seq. | Data Throughput/Minute |
|---|---|---|---|
| w/o Packing | 208,334 | 1 | 169.82k |
| w/ Packing (Ours) | 47,350 | 4.4 | 739.95 |

Table 11: Ablation study of masking method.

| | Fine-tune | | Linear-prob | |
|---|---|---|---|---|
| Mask Method | RML2016.10B | EM-Infer-Comm | RML2016.10B | EM-Infer-Comm |
| Per-sequence | 56.40 | 81.64 | 38.75 | 54.25 |
| Per-sample (Ours) | 57.51 | 83.11 | 44.50 | 58.07 |

**Mask Method.** We conducted the ablation study on masking methods, comparing per-sample masking and per-sequence masking. The strictly controlled per-sample masking ensured that the mask ratio was precisely enforced, leading to better performance, as shown in Table 11.

**Mask Ratio.** The ablation study on mask ratios shows that 75% mask ratio achieves the best performance, which is adopted in our proposed EMind, as presented in Table 12.

Table 12: Ablation study of masking ratio.

| | Fine-tune | | Linear-prob | |
|---|---|---|---|---|
| Mask ratio | RML2016.10B | EM-Infer-Comm | RML2016.10B | EM-Infer-Comm |
| 20% | 52.98 | 79.77 | 40.56 | 56.72 |
| 50% | 54.42 | 81.16 | 43.23 | 55.46 |
| 75% (Ours) | 57.51 | 83.11 | 44.50 | 58.07 |
| 90% | 53.88 | 80.85 | 42.99 | 55.85 |

Table 13: Ablation study of max sequence length.

| | Fine-tune | | Linear-prob | |
|---|---|---|---|---|
| Max Seq. Length | RML2016.10B | EM-Infer-Comm | RML2016.10B | EM-Infer-Comm |
| 2048 | 56.94 | 81.87 | 40.30 | 56.05 |
| 6000 (Ours) | 57.51 | 83.11 | 44.50 | 58.07 |
| 9000 | 56.51 | 81.34 | 39.93 | 56.92 |

**Max Sequence Length.** We further conducted an ablation study on the setting of max sequence length, using 2,048, 6,000, and 9,000. The results show that the length of 6,000 outperforms the others, as shown in Table 13.

# 6 RELATED WORKS

**Electromagnetic signal Datasets.** While scalable, simulated data may lack real-world fidelity (Zhang et al., 2019; Guler et al., 2025). Thus, recent work incorporates real-world samples: Aboulfotouh et al. (2025) pre-trained on RF spectrograms and WiFi/5G CSI from diverse locations, and Zhou et al. (2025) compiled a dataset from public sources (RML2018.01A) and practical scenarios (O'Shea et al., 2018; Fontaine et al., 2019). Despite the trend toward more realistic data, existing datasets often lack the diversity and scale needed to improve FMs' generalization (Aboulfotouh et al., 2025).

**Electromagnetic signal Foundation Models.** The design of EM foundation models is crucial for capturing complex correlations in wireless data, with many adopting Transformer-based architectures inspired by advances in NLP and CV (Awais et al., 2025; Han et al., 2024; Hong et al., 2023; Vaswani et al., 2017; Yang et al., 2025; Guler et al., 2025). For instance, Yang et al. (2025) modeled spatial, temporal, and frequency correlations with Transformers, while Aboulfotouh et al. (2025) employed ViTs, and Zhou et al. (2025) proposed a CNN–self-attention hybrid. Beyond architecture, diverse pre-training strategies have been explored: masked signal modeling (Aboulfotouh et al., 2025; Guler et al., 2025), sometimes enhanced with contrastive learning (Guler et al., 2025), and hybrid objectives such as masked reconstruction with next-slot prediction (Zhou et al., 2025). However, these methods are often designed for specific data formats, limiting efficiency across heterogeneous datasets.

# 7 CONCLUSION

In this paper, we propose a new pre-training pipeline for EM signals, including a large-scale dataset and a specialized IQ-based architecture. We build EMdata-81M, a raw IQ dataset for large-scale pre-training, offering broader scale, richer scenarios, and more diverse signals with fine-grained physical attributes relevant to downstream tasks. Benchmarking EMdata-81M with our foundation model EMind demonstrates clear advantages in EM signal understanding. Experiments show that EMind improves modulation classification, parameter regression, blind source separation, and denoising, while its features generalize well across tasks. Future work will focus on refining training strategies and standardizing EM signal attributes to further boost generalization and accuracy.

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
