# EMIND: A FOUNDATION MODEL FOR MULTI-TASK ELECTROMAGNETIC SIGNALS UNDERSTANDING

## A APPENDIX

### A.1 OVERVIEW

This material provides additional details about the proposed EMind and EMdata-81M, as well as experimental settings and results not included in the main text due to page limit. Its organization is as follows:

Sec. A.2 visualizes the samples of EMdata-81M and category and annotation details.

Sec. A.3 provides model architecture details of EMind.

Sec. A.4 provides the full experiment configurations of pretraining and downstream tasks.

Sec. A.5 provides more visualizationresults and impldent details of BSS and SD.

### A.2 MORE DETAILS ON EMDATA-81M

#### A.2.1 SAMPLE VISUALIZATIONS

EM-Comm

HisarMod2019.1

Panoradio HF

RadarCommDataset

EM-RadarParaIQSim

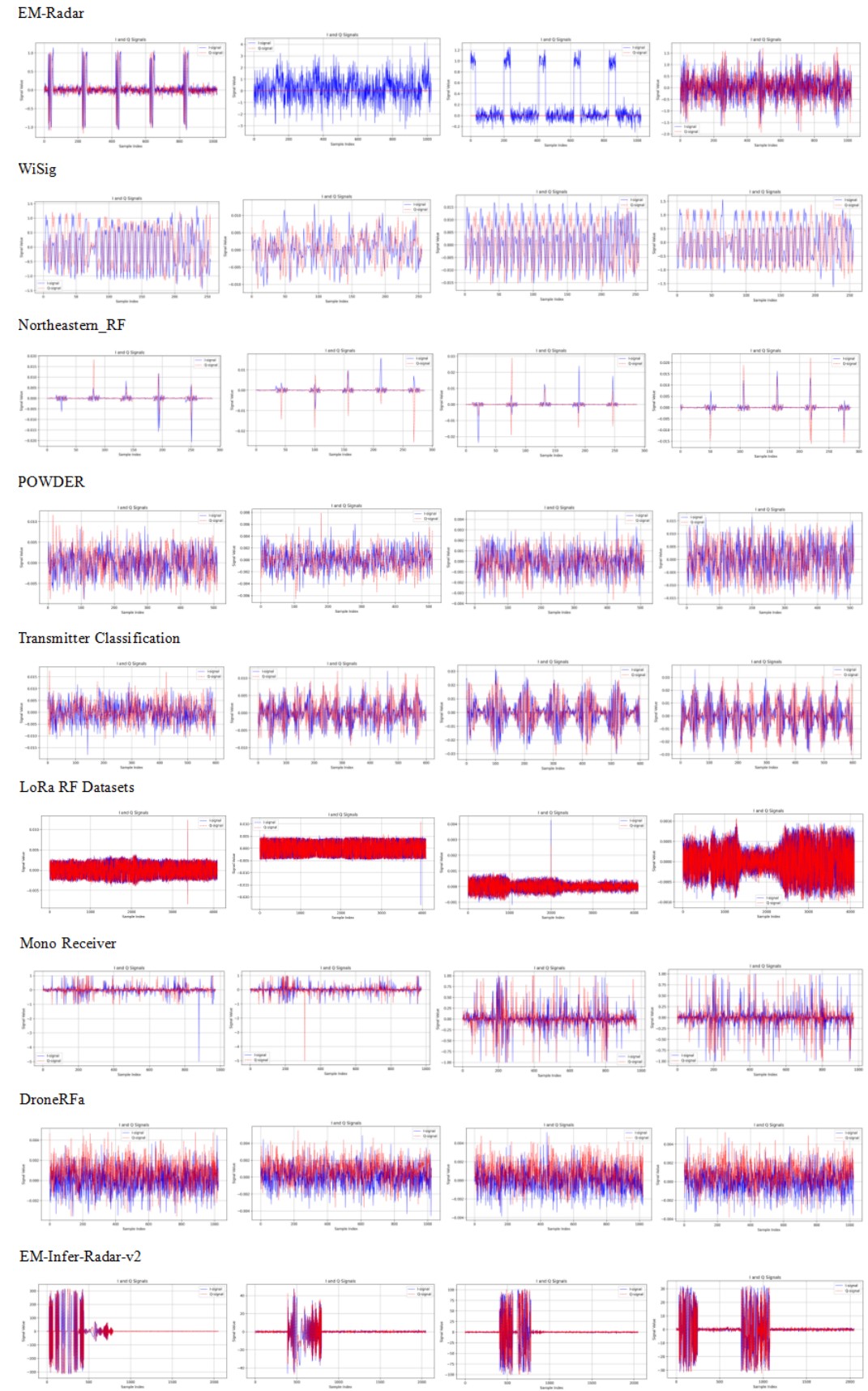

Figure 1: Visualization of signal samples of EMdata-81M, four samples are randomly drawn from each dataset.

### A.2.2 CATEGORY DETAILS

EMdata-81M stands out in terms of scene types, signal lengths, and dataset scale, as shown in Figure 2

Specifically, EMdata-81M encompasses **four scene types**: communication, radar, RF, and interference.

**Eight signal types** are included: airborne detection radar, airborne range radar, air-ground MTI radar, ground mapping radar, radar altimeter, SATCOM, AM radio, and short-range wireless.

**Twenty-nine modulation schemes** are covered: BPSK, QPSK, 8PSK, 16PSK, 32PSK, 64PSK, 4QAM, 8QAM, 16QAM, 32QAM, 64QAM, 128QAM, 256QAM, 2FSK, 4FSK, 8FSK, 16FSK, 4PAM, 8PAM, 16PAM, AM-DSB, AM-DSB-SC, AM-USB, AM-LSB, FM, PM, MORSE, PSK31, and PSK63.

**Nine radar waveforms** are included: coherent pulse train, barker code, polyphase barker code, frank code, linear frequency modulated (LFM), rectangular, phase coded, stepped FM, and custom FM.

**Twelve interference types** are also included: pure noise, intermittent sampling forwarding interference, spot-jamming interference, blocking interference, frequency-sweeping interference, range-fooling interference, dense false target interference, smart noise interference, chaff interference, chaff interference combined with intermittent sampling forwarding interference, dense false target interference combined with smart noise interference, and range-fooling interference combined with FM-sweeping interference.

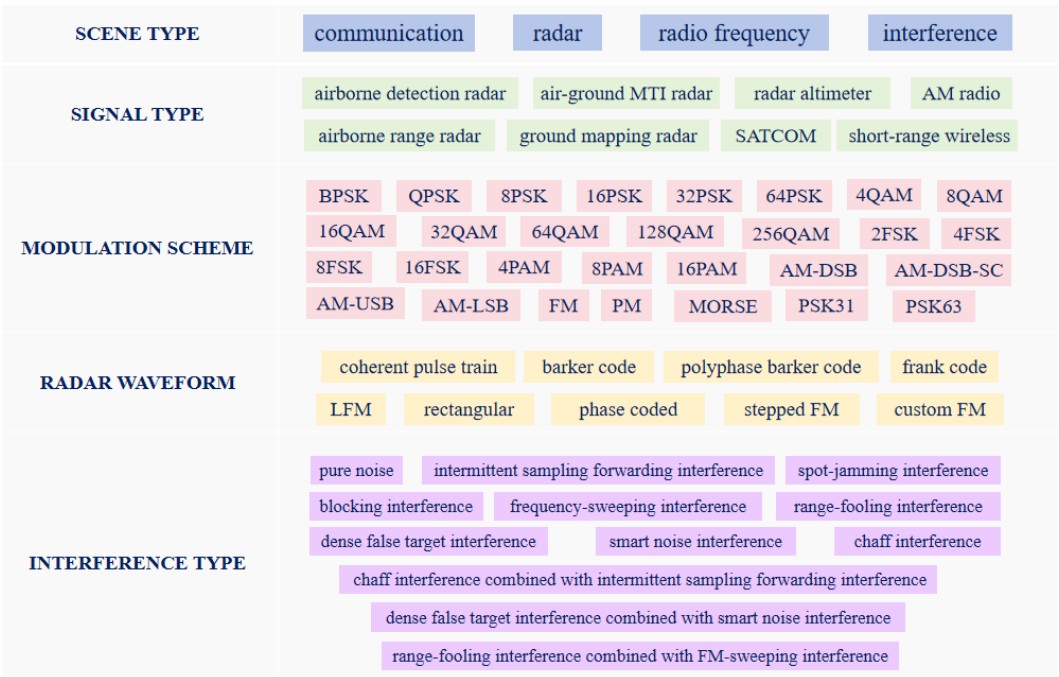

Figure 2: Category Details of EMdata-81M.

### A.2.3 ANNOTATION DETAILS

We standardize the annotations for signal attributes and extract the following 17 key attributes:

```
iq_data, dataset_name, sampling_rate, device_id, transmission_id,
infer_class,  snr,  isr,  modulation_type,  radar_waveform_type,
pri, pulse_time_delay, number_of_pulses, pulse_width, bandwidth,
amplitude, radar_segmentation_type
```

The attribute is set to None if any of these fields are missing in a specific dataset.

### A.3 MORE DETAILS OF EMIND ARCHITECTURE

The sampling rate is a special attribute of electromagnetic signals, varying widely across different datasets with a dynamic range from kilohertz to megahertz. We dynamically insert a "sampling rate token" for each signal sample. We modify the network architecture so that the sampling rate token is excluded from random masking but always participates in training, enabling the model to naturally acquire the sampling rate information corresponding to the current signal and allowing tasks sensitive to sampling rate, such as pulse width, pulse repetition interval (PRI), and bandwidth, to be properly aligned.

### A.4 CONFIGURATIONS OF PRE-TRAINING AND FINE-TUNING

#### A.4.1 PRE-TRAINING SETUP

Two versions of the foundation model, EMind-Base and EMind-Large are released, using data with a signal-to-noise ratio (SNR) greater than 6 for pretraining whenever available. EMind-Base consists of a 12-layer encoder and an 8-layer decoder, with a mask ratio of 75%, a patch size of 8, a maximum sequence length of 6,000, and employs the eager attention mechanism. It is trained for 10 epochs with a batch size of 40, using the AdamW optimizer with a base learning rate of $1e^{-4}$ and a warm-up ratio of 10%, resulting in a total of 85.88 million parameters. EMind-Large increases the encoder depth to 24 layers, resulting in a total of 303.67 million parameters.

#### A.4.2 DATASET SAMPLING WEIGHTS

To enhance the model's adaptability across different datasets, we propose an adjustable dataset weighting sampling mechanism. This mechanism reduces the sampling ratio for datasets prone to overfitting and increases it for datasets that are difficult to learn, thereby improving the model's ability to capture features from challenging subsets during training and enhancing overall generalization performance. Ultimately, the sampling ratios for our pretraining datasets are set as follows:

| | | | |
|---|---|---|---|
| EM-Comm | 1 | HisarMod2019.1 | 0.5 |
| Panoradio HF | 1 | RadarCommDataset | 1 |
| EM-RadarParaIQSim | 0.5 | EM-Radar | 0.5 |
| WiSig | 1 | Northeastern RF | 1 |
| POWDER | 1 | Transmitter Classification | 1 |
| LoRa RF Datasets | 1 | Mono Receiver | 1 |
| DroneRFa | 0.5 | EM-Infer-Radar-v2 | 0.5 |

The final pretrained weight is taken from the last iteration of epoch 3.6.

#### A.4.3 SIGNAL DATA PREPROCESSING

The IQ data are normalized using absolute magnitude normalization. Considering the physical characteristics of EM signals, the magnitude often directly reflects the signal's strength and energy distribution. Applying absolute magnitude normalization helps preserve this critical feature, to enhance the model's ability to perceive the physical properties of the signal, which is

$$\hat{\mathbf{IQ}} = \mathbf{IQ}/(\max(|\mathbf{IQ}|)). \tag{1}$$

The regression parameters are also normalized, particularly when dealing with small time values (such as radar parameters in $\mu$s), to improve the training performance and convergence of regression tasks. For radar parameters, min-max normalization is applied.

#### A.4.4 MORE DETAILS OF DOWNSTREAM TASK SETUPS

For the few-shot experiments on RML2016.10A, we followed the few-shot experiment settings by randomly selecting 50 or 100 samples per modulation class and SNR level from the training set to form the support set, with the remaining samples used for validation and testing. Similarly, for RadChar, experiments are conducted under 10-shot, 50-shot, and 100-shot settings, as shown in Table 1. The table also provides configuration details of fine-tuning experiments.

Table 1: Downstream Dataset Setting and Splitting Strategy.

| TASK | DATASET | ATTRIBUTE | SIGNAL LENGTH | FINE-TUNE SETTING | FEW-SHOT SETTING |
|------|---------|-----------|--------------|-------------------|------------------|
| CLASSIFICATION | RML2016.10A | modulation | 128 | 1:9 | 50 or 100 samples/class/SNR |
| | RML2016.10B | modulation | 128 | 1:9 | |
| | RML2016.10C | modulation | 128 | 1:9 | |
| | RML2018.01A | modulation | 1,024 | 1:9 | |
| | ADS-B | device id | 3,000 | 1:9 | |
| | EM-AIS | device id | 3,840 | 5:5 | |
| | EM-Infer-Comm | infer class | 1,024 | 1:9 | |
| | EM-Infer-Radar | infer class | 1,024 | 1:9 | |
| REGRESSION | RadChar | radar waveform type pulse repetition interval (PRI) pluse time delay number of pulses pluse width | 512 | 70:15 | 10 or 50 or 100 samples/class/SNR |
| RECONSTRUCTION | EM-Radar-Mix | iq data | 1,024 | 8:1 | |
| | EM-Signal-Denoise | iq data | 1,024 | 8:1 | |

### A.4.5 MORE DETAILS OF DOWNSTREAM DATASETS

This section provides an overview of the self-curated downstream datasets.

**EM-AIS** dataset is collected using a high-frequency receiver. This dataset uses the Maritime Mobile Service Identity (MMSI) as a unique device identifier (device_id) for vessel identification. MMSI consists of 9 digits and can uniquely identify ships, offshore facilities, and shore-based radio stations. A total of 12,531 samples are collected. Since the original EM-AIS signal sequences are relatively long (11,520), we applied threefold downsampling based on their bandwidth characteristics and the Nyquist sampling criterion, resulting in a final signal length of 3,840.

The interference dataset is synthetically generated and consists of two parts: the communication interference dataset EM-Infer-Comm and the radar interference dataset EM-Infer-Radar. It is designed to provide diverse training data for evaluating the performance of communication and radar systems in interference environments. **EM-Infer-Comm** includes five types of communication signals, with different modulation schemes set for each signal type. In addition, the dataset covers nine types of interference, which are simulated under various real-world complex conditions using different parameters such as interference duration, frequency, and bandwidth. **EM-Infer-Radar** includes five types of radar signal schemes, simulating the various signal forms that a radar system may encounter in real-world applications. To ensure data diversity, the dataset also incorporates twelve types of interference, enabling a comprehensive evaluation of radar system performance under interference conditions.

**EM-Radar-Mix** is a simulated blind source separation dataset, consisting of eight types of radar signals with a SNR of 12 dB as source signals. When constructing samples, these signals are combined in groups of two or fewer, and the specific signal types mixed in each sample are unknown to the model, thus forming a typical blind source separation scenario. In this task, the model is required to recover the original components from the mixed signals without prior knowledge of the number of sources, signal constituents, or their structure.

**EM-Denoise-Signal** is a high-fidelity simulated dataset specifically constructed for electromagnetic signal denoising tasks. It covers typical protocols in both communication and radar domains, with 20 modulation schemes. All samples are corrupted with additive white Gaussian noise (AWGN), with a signal-to-noise ratio (SNR) ranging from –3 dB to 20 dB, and additionally incorporate system frequency offsets (±50 kHz) as well as IQ amplitude and phase imbalance modeling.

### A.4.6 METRICS

To systematically and comprehensively evaluate the model's performance, we introduce multiple evaluation metrics covering quantitative analysis across different task dimensions, including classification, regression, blind source separation and signal denoise.

In classification tasks, Overall Accuracy (OA) is used to measure the overall classification performance. OA represents the proportion of correctly predicted samples to the total number of samples, reflecting the model's average classification accuracy across all test samples, which is,

$$OA = \frac{\sum_{i=1}^{k} n_{ii}}{N}, \tag{2}$$

where $n_{ii}$ denotes the number of samples correctly classified for class $i$, $k$ is the total number of classes, $N$ is the total number of samples. OA ranges from 0 to 1, with higher values indicating better classification performance.

In regression tasks, Mean Absolute Error (MAE) is used as a evluation metric. MAE is defined as the average of the absolute differences between the predicted values and the true values, effectively measuring the deviation of the model's predictions from the ground truth, which is,

$$\text{MAE} = \frac{1}{N} \sum_{i=1}^{N} |y_i - \hat{y}_i|, \tag{3}$$

where $y_i$ is the true value, $\hat{y}_i$ is the predicted value, and $N$ is the total number of samples.

In the blind source separation task, we evaluate performance on 100 randomly selected test samples. The metrics include signal-to-distortion ratio (SDR), signal-to-interference ratio (SIR), signal-to-artifact ratio (SAR), and scale-invariant SDR (SI-SDR), and mean squared error (MSE). Specifically, SDR quantifies the ratio of the target signal power to all distortion components (including noise, interference, and artifacts), representing the overall separation quality, which is,

$$\text{SDR} = 10 \log_{10} \frac{\|s_{\text{target}}\|^2}{\|e_{\text{total}}\|^2}. \tag{4}$$

SIR reflects the model's ability to suppress interference from other sources:

$$\text{SIR} = 10 \log_{10} \frac{\|s_{\text{target}}\|^2}{\|e_{\text{interf}}\|^2}. \tag{5}$$

SAR quantifies the amount of artifacts introduced during separation, indicating the proportion of spurious components:

$$\text{SAR} = 10 \log_{10} \frac{\|s_{\text{target}} + e_{\text{interf}}\|^2}{\|e_{\text{artif}}\|^2}. \tag{6}$$

SI-SDR removes the influence of gain in SDR to provide a more robust evaluation of signal quality:

$$\text{SI} - \text{SDR} = 10 \log_{10} \frac{\|\alpha s\|^2}{\|\alpha s - \hat{s}\|^2}, \quad \alpha = \frac{\hat{s}^\top s}{\|s\|^2}. \tag{7}$$

MSE measures the overall reconstruction accuracy of the separated signal relative to the reference signal in the time domain, which is,

$$\text{MSE} = \frac{1}{N} \sum_{i=1}^{N} (s_i - \hat{s}_i)^2, \tag{8}$$

where $s_i$ is the reference signal and $\hat{s}_i$ is the estimated signal. For the denoising task, we adopt MSE as the evaluation metric, as well.

In few-shot classification tasks, Cohen's Kappa coefficient (Kappa) is further introduced. It is a statistical measure used to assess inter-rater agreement or classification reliability, taking into account the possibility of agreement occurring by chance, serving as a more stringent metric for evaluating overall classification consistency. Kappa is defined as:

$$\text{Kappa} = \frac{\text{OA} - P_E}{1 - P_E}, \tag{9}$$

where OA is the observed accuracy, and $P_E$ is the expected agreement by chance, calculated as:

$$P_E = \sum_{i=1}^{k} \frac{\left( \sum_{j=1}^{k} n_{ij} \cdot \sum_{j=1}^{k} n_{ji} \right)}{N^2}, \tag{10}$$

where $n_{ij}$ denotes the number of samples whose true class is $i$ but are classified as class $j$. The value of Kappa ranges from -1 to 1, where Kappa = 1 indicates perfect agreement, Kappa = 0 indicates agreement equivalent to chance, and Kappa ¡ 0 indicates agreement worse than chance.Together with other metrics, it provides a comprehensive quantitative basis for evaluating model performance across multi-task and multi-dimensional scenarios.

### A.5 More Experiments

#### A.5.1 More details of Blind source separation

To acieve blind source separation, a fine-tuning framework based on an autoencoder (AE) is designed. The AE architecture is first initialized with the identifiable latent variables obtained from the pre-trained masked autoencoder (MAE) reconstruction task model, thereby preserving high-level semantic information. Multiple linear compression layers are then introduced on top of the pre-trained model, and the decoder capacity is restricted to map the mixed signal into a low-dimensional latent space, forcing the model to extract concise and discriminative latent features. As shown in Table 2, the linear compression layers are applied layer-by-layer according to the preset hidden dimensions, and a final linear projection compresses the features into a fixed number of $K$ channels ($K = 2$ in our setting), each represented by a 16-dimensional vector, to reduce the signal representation complexity. During training, the loss function comprises a reconstruction loss and an $\ell_2$ regularization term on the latent representation. The reconstruction loss can be applied to either the mixed signals or to each individual separated prediction, with individual reconstruction loss resolved by a permutation-invariant training strategy to handle signal order ambiguity. The $\ell_2$ regularization encourages the prediction to remain stable, sparse, and discriminative. This MAE semantic initialization and AE fine-tuning framework retains the semantics learned by MAE and enables effective separation of latent source signals under unsupervised conditions.

Table 2: Model layers architecture for the downstream task of blind source separation. Note: K = 2.

| INPUT | Layer 1 | Layer 2 | Layer 3 | Layer 4 | OUTPUT |
|---|---|---|---|---|---|
| $D_{Enc} \times num_{patch}$ | 4096 | 2048 | 1536 | 1024 | $16K$ |

Figure 3 shows the visualization results of the blind source separation task. Figure 3 (a) is the input mixed IQ signals, Figure 3 (b) shows the separation results from training from scratch, Figure 3 (c) shows the separation results after fine-tuning by loading the pre-trained model, and (d) is the ground truth. From these visualizations, we can intuitively observe the profound impact of pre-trained model on performance. After loading the pre-trained model, the model quickly converges to the ideal separation results, successfully distinguishing signals from different sources, and performing excellently in terms of signal integrity and accuracy. In contrast, without pre-trained weights, the solution to this ill-posed problem tends to be infinitely many, and unsupervised training struggles to obtain a sufficiently general representation of IQ signals. The model faces significant difficulties in signal separation, unable to effectively extract the key features of the signals, ultimately leading to separation failure.

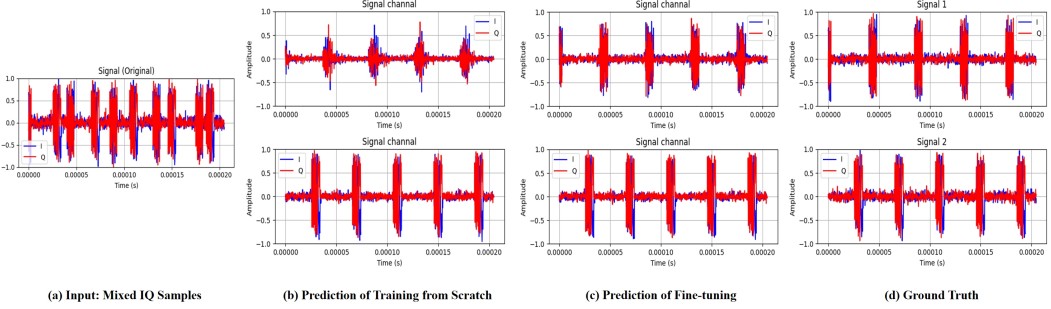

(a) Input: Mixed IQ Samples     (b) Prediction of Training from Scratch     (c) Prediction of Fine-tuning     (d) Ground Truth

Figure 3: Visualization of signal blind source separation (BSS) results. (a) mixed signal input, (b) prediction of training from scratch, (c) prediction of fine-tuning, (d) ground truth. The comparative results demonstrate that the model is effective in the blind source separation task, achieving good consistency in both amplitude and phase alignment.

#### A.5.2 Visualization of Signal Denoise

For our model framework, the input consists of noisy signals, with no denoised signals available as supervision. Using an autoencoder, the model is able to separate the true signal from the noisy input.

Further visualization results are shown in Fig.4. Under noisy input (Noise IQ) conditions, the model initialized with pre-trained weights significantly outperforms training from scratch, with its denoised predictions being more consistent with the ground truth in terms of both amplitude and phase.

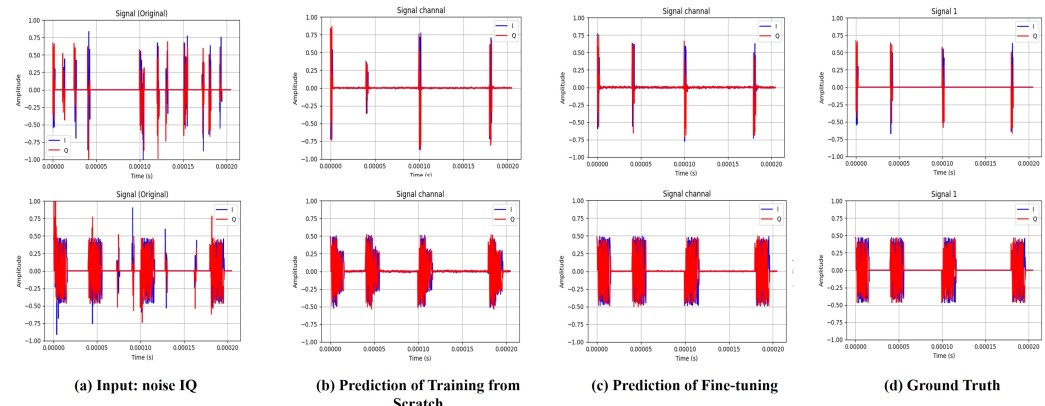

(a) Input: noise IQ    (b) Prediction of Training from Scratch    (c) Prediction of Fine-tuning    (d) Ground Truth

Figure 4: Visualization of IQ signal denoising. (a) noise IQ input, (b) prediction of training from scratch, (c) prediction of fine-tuning, (d) ground truth. The comparison illustrates the model's effectiveness in denoising, achieving consistency in amplitude and phase alignment.