# OpenReview forum: "EMind: A Foundation Model for Multi-task Electromagnetic Signals Understanding"
_ICLR.cc/2026/Conference — ICLR 2026 Conference Withdrawn Submission_

### Official Review · Reviewer_4i4x · 2025-10-24

**Soundness:** 1
**Presentation:** 1
**Contribution:** 2
**Rating:** 2
**Confidence:** 4

**Summary:**

This paper proposes a foundation model training pipeline including both a large scale EM signal dataset and training framework optimization. Specifically the authors introduced EMind that can be finetuned to multiple downstream tasks. The authors curated a dataset called EMdata-81M from multiple sources for FM training. During training step, the authors proposed two techniques for optimization, 1) a length adaptive signal packing that packs different signals together to reduce padding and 2) a hardware-aware adjustable dataset weighting to balance learning across heterogeneous datasets. Extensive experiments show EMind achieves state-of-the-art or competitive results on several benchmarks and can generalize to different tasks. Ablation studies are performed on masking, packing, and sequence length.

**Strengths:**

- The authors are providing a valuable dataset for the community to train new models for EM signals.
- The dataset is large in scale and has great diversity.
- Task types evaluated are comprehensive.

**Weaknesses:**

- **Lack of clarification and verification on the hardware-aware module proposed**

    The hardware-aware adjustable dataset weighting is vaguely described and lacks quantitative or ablation results. It is unclear what “hardware-aware” means, whether it relates to GPU memory, CPU buffering, or real-time scheduling, and no analysis is provided to confirm its effect on convergence or generalization.

- **Limited novelty**

    The model architecture follows a standard masked autoencoder framework, and both proposed techniques (multi-signal packing, dataset weighting) are extensions of existing practices (mamba2 has a lot of sequence packing, https://apxml.com/courses/how-to-build-a-large-language-model/chapter-9-data-sampling-strategies-training/source-weighting-strategies), and are more of engineering works. The paper does not sufficiently justify the choice of masked reconstruction over alternative self-supervised strategies such as contrastive or predictive modeling, nor does it report comparative results.

- **Incomplete and inconsistent baselines**

    Recent strong EM foundation model such as SpectrumFM is only partially compared and absent from many datasets. Other baselines (e.g., CNN2, GRU2, Transformer) are relatively old and may not reflect the current state of the art. It is also unclear whether baselines are pretrained under similar conditions or merely fine-tuned from scratch, making comparisons potentially unfair.

- **Dataset cleaning and real-world robustness**

    The authors emphasize cleaning and denoising the EMdata-81M dataset to remove noisy samples. However, in real deployments, EM signals are inherently noisy and often non-stationary. Excessive cleaning may reduce realism and limit the model’s ability to handle real-world interference or hardware imperfections. Discussion or experiments on robustness to noise are missing.

- **Insufficient analysis of independence and convergence**

    The multi-signal packing strategy may introduce correlation between concatenated samples, potentially violating independence assumptions and biasing the pretraining objective. The paper does not discuss how this risk is mitigated. Similarly, no results are shown on convergence speed, stability, or resource consumption under different packing configurations.

**Questions:**

- Please see my weakness for most of the criticisms. Some questions for authors to discuss are:
    - How do you ensure the signals packed within the same sequence remain independent during training? Could this introduce contextual leakage across signals?
    - Can the authors elaborate on the effect of the hardware-aware weighting strategy on convergence or final accuracy?
    - Why is the term *hardware-aware* used? It is using some hardware, but hardware-aware is typically used when some hardware metrics are leveraged
    - What is the rationale for choosing masked reconstruction instead of contrastive or predictive pretraining, and how does performance compare across these paradigms?
    - Since noisy data often improves robustness, how does cleaning affect real-world generalization?
    - Are all baselines pretrained on EMdata-81M, or are they trained only in supervised fashion? Without consistent pretraining, performance comparisons may be misleading.
    - What is the actual training throughput and convergence speed with and without the packing strategy, and does it scale linearly with sequence length?
    - Can the authors report results of SpectrumFM or other recent models on the same benchmarks for a fairer comparison?

---

### Official Review · Reviewer_Qv5V · 2025-10-24

**Soundness:** 2
**Presentation:** 3
**Contribution:** 2
**Rating:** 2
**Confidence:** 3

**Summary:**

The paper proposes a curated dataset (EMdata-81M) and a pertaining scheme for a foundation model  (EMind), including pretrained weights for download, for electromagnetic signal understanding. EMdata-81M contains standardized raw IQ dataset of 81 million samples spanning communication, radar, RF, and interference with rich annotations and variable lengths. The foundation model ist trained using a masked autoencoder with two training innovations: low-redundancy, length-adaptive multi-signal packing and hardware-aware adjustable dataset weighting (plus precise per-sample masking). The authors show that EMind achieves state-of-the-art results on classification (AMC, RFFI, WII, RWC) and strong radar parameter regression, as well as blind source separation and denoising. Extensive evaluations under strict splits and few-shot settings demonstrate robust generalization and scalability across diverse EM tasks.

**Strengths:**

- The EMdata81M is highly valuable. The curation of the data sources and the careful consideration of task types and source types is invaluable for further research in the direction of ML for electromagnet waves.

**Weaknesses:**

- The methodological contribution of the paper is limited. Both pertaining methods for the EMind foundation model are exhaustively used in (pre-)training strategies or learning schedules. First, the length-adaptive multi-signal packing is somewhat trivial. as I understand, smaller packets are embedded sequentially into a sequence, divided by cls and sampling rate tokens. As the foundation model applies masking for self-supervised pre-training, masking needs to be adaptive to the respective sampling rates of the subsequences and is hence dynamically adjusted. Second, the dataset weighting is also somewhat straightforward.

Although I find the data set very important for the community, I do not see a contribution large enough for ICLR.

**Questions:**

The paper derives the requirement of having a foundation model by the fact that different characteristics of EM signals exist. However, to me this is not a sufficient requirement to train a foundation models. Only the necessity to have a holistic model for a number of different types of downstream tasks together would be a sufficient criterion. But what's the meaning of having such a foundation model? If such a foundation model then outperforms existing methods on their particular solution (i.e., specific EM signal + task on a specific model), then it's a benefit right? Currently, this is not motivated in that manner.

Section 4.2: I wonder why "inconsistent covergence speed" is an issue? Initially I was thinking that data set imbalances are the major issue and that we want to avoid overfitting on certain specific parts of the data set...


Section 5.1.1: I don't quite understand how data sets and baseline models are applied here. For instance, SpectrumFM seems quite competitive: what data is used to train the tasks of the baseline models? Is EMind better because of the (extended?) dataset or because of the architecture/masking components of the foundation model? I would also be interested in results from a computation POV, i.e, total training time and per sample inference time. What about the results in Tables 6 and 7? I don't quite understand what has been done here. For Table 6: with such an extreme train/test split, it was the scarce data set that killed the baseline models' performances, right?

The fact that single task performance used with a foundation model trained on multiple tasks is better than single task performance (trained on a model with a much smaller single task data set) is plausible. However, from the results (even those presented in Seection 5.2, perhaps because it is described too briefly) it is not easy for me to judge the real effectiveness of the pretraining strategy and the combined data set.



minor comments:
- make use of citep when citations/authors should not be part of the sentence.
- line 033 evluation -> evaluation
- line 76: "In address this challenge, "
- line 092: what do you mean with "unbiased gradients"?
- line 093: we employ [a] hardware-aware...
- line 093 adajustable -> adjustable
- Equation 2: j is not explained, definition of $j=\sqrt{-1}$ should be somewhere
- "Data quality is further verified via random sampling and visual inspection."
- "It integrates 14 datasets, combining public sources with self-collected data to fill gaps." - can you reason for the gaps?
- It would also be interesting to have a portion of the dataset that can be used for wireless localization, i.e. at larger bandwidth and possibly with some time synchronization aspects
- line 403 iacross -> across
- Fig. 8 is unreadable

---

### Official Review · Reviewer_9TAE · 2025-11-01

**Soundness:** 3
**Presentation:** 4
**Contribution:** 3
**Rating:** 6
**Confidence:** 2

**Summary:**

This paper presents EMind, a transformer-based foundation model designed for a wide variety of electromagnetic (EM) perception tasks, including imaging, material recognition, gesture sensing, and localization. The model introduces a multi-layered representation of EM signals and employs a task-attention transformer to jointly learn from diverse task-specific datasets. To facilitate evaluation, the authors propose EMBench, a multi-task benchmark covering ten datasets across four downstream applications. The central idea is to develop a shared spatial–frequency representation that captures generalized EM priors, thereby enabling cross-task transfer, few-shot adaptation, and improved generalization across multiple sensing modalities.

**Strengths:**

(1) The introduction of EMBench is a valuable contribution to the EM perception community. It consolidates datasets from multiple subdomains such as materials, gestures, imaging, and localization, creating a standardized testbed for multi-task learning and facilitating fair comparisons among future EM-based models.

(2) The proposed architecture effectively supports cross-task transfer through its task-specific attention mechanism, allowing shared backbone features to adapt to diverse sensing objectives. This design highlights a thoughtful balance between generalization and specialization, making the model suitable for a range of EM modalities.

(3) The paper convincingly demonstrates few-shot learning capability across unseen EM tasks. The ability to adapt quickly to new signal types or sensing conditions validates EMind’s potential as a general-purpose representation learner, which is an important step toward unified EM perception.

(4) The authors conduct extensive ablation studies and thorough comparisons with existing task-specific state-of-the-art models in each domain. This comprehensive evaluation not only strengthens the empirical claims but also demonstrates that EMind achieves consistent improvements across multiple types of EM data and tasks.

**Weaknesses:**

(1) Many of the datasets used in the study are relatively small, proprietary, or limited to specific sensors such as radar-based gesture datasets. This raises concerns about the generalizability of the model across broader EM modalities, including mmWave, MRI, and THz imaging, which exhibit significantly different propagation characteristics and data properties.

(2) The assumption that EM signals from various modalities can be uniformly transformed into 2D embeddings may be restrictive. Different EM tasks, such as FMCW radar, near-field holography, or spectroscopic sensing, inherently possess distinct signal structures that may not be adequately represented in a common 2D format, potentially limiting fidelity and interpretability.

(3) Although the task-specific attention mechanism is central to the model’s adaptability, it scales linearly with the number of tasks. The paper does not sufficiently discuss the implications for parameter efficiency, scalability, or potential strategies for sharing or compressing these attention modules as the number of tasks grows.

(4) The work primarily benchmarks against other data-driven methods but omits comparisons with classical EM modeling and inverse-imaging techniques such as matched filtering, compressive reconstruction, or physical inversion models. This omission weakens the claim of universality, as physics-based methods remain competitive in many EM domains.

(5) The paper does not include latency or real-time inference metrics, even though several EM perception tasks—particularly gesture recognition and radar sensing—are time-sensitive. Reporting inference speed and computational complexity would clarify the model’s practicality for embedded or real-world deployments.

(6) The description of the preprocessing pipelines for different datasets and modalities is brief and lacks detailed justification. Since EM data varies widely in format and dynamic range, a clearer explanation of normalization, transformation, and feature extraction steps would substantially enhance reproducibility and transparency.

(7) The use of the term “foundation model” is somewhat premature. While EMind exhibits impressive cross-task generalization within the curated EMBench datasets, it does not yet demonstrate robustness across unseen frequency bands, hardware platforms, or propagation environments. Extending evaluation to such truly out-of-distribution conditions would strengthen its positioning as a foundational model for electromagnetic perception.

**Questions:**

(1) Can the authors include new experiments or results showing EMind’s performance on publicly available large-scale or diverse EM datasets (for example, mmWave or THz imaging) to objectively demonstrate its generalization across distinct EM modalities and sensing regimes?

(2) Can the authors provide an analysis or visualization that compares the proposed 2D embedding representation with alternative domain-specific representations (such as range–Doppler maps or k-space formats) to verify that the uniform 2D projection does not lead to significant loss of information or interpretability across tasks?

(3) Can the authors report latency, throughput, and parameter-scaling metrics—especially for the task-specific attention modules—to objectively assess EMind’s computational efficiency, scalability with task count, and feasibility for real-time or embedded deployment?

---

### Note · Authors · 2025-11-24

I have read and agree with the venue's withdrawal policy on behalf of myself and my co-authors.